# Clinical Outcome and Prognostic Factors of Pancreatic Adenosquamous Carcinoma Compared to Ductal Adenocarcinoma—Results from the German Cancer Registry Group

**DOI:** 10.3390/cancers14163946

**Published:** 2022-08-16

**Authors:** Rüdiger Braun, Monika Klinkhammer-Schalke, Sylke Ruth Zeissig, Kees Kleihus van Tol, Louisa Bolm, Kim C. Honselmann, Ekaterina Petrova, Hryhoriy Lapshyn, Steffen Deichmann, Thaer S. A. Abdalla, Benjamin Heckelmann, Peter Bronsert, Sergii Zemskov, Richard Hummel, Tobias Keck, Ulrich F. Wellner

**Affiliations:** 1Department of Surgery, University Medical Center Schleswig-Holstein, Campus Lübeck, 23562 Lübeck, Germany; 2Network for Care, Quality and Research in Oncology (ADT), German Cancer Registry Group of the Society of German Tumor Centers, 14057 Berlin, Germany; 3Institute for Surgical Pathology, Faculty of Medicine, Medical Center, University of Freiburg, 79085 Freiburg, Germany; 4Department of General Surgery, Bogomolets National Medical University, 01601 Kiev, Ukraine

**Keywords:** pancreatic cancer, adenosquamous carcinoma, ductal adenocarcinoma, overall survival

## Abstract

**Simple Summary:**

About 0.5% of pancreatic malignancies are adenosquamous carcinomas. The pathophysiology of these carcinomas is poorly understood and clinical data is sparse. We compared characteristics and prognostic factors of adenosquamous carcinoma and ductal adenocarcinoma, which represent the most common type of pancreatic cancer based on data from the German cancer registry group. Adenosquamous carcinoma showed poorer differentiation and higher frequency of blood vessel invasion indicative of a more aggressive tumor biology. Adenosquamous tumor differentiation was a strong negative prognostic factor. Survival of patients with adenosquamous carcinoma was shorter compared to patients with a ductal adenocarcinoma after surgical tumor resection. This study suggests that distinct multimodal treatment protocols should be considered for adenosquamous carcinomas of the pancreas.

**Abstract:**

Background: Adenosquamous carcinoma of the pancreas (ASCP) is a rare malignancy and its pathophysiology is poorly understood. Sparse clinical data suggest that clinical outcome and overall survival is worse in comparison to common pancreatic ductal adenocarcinoma (PDAC). Methods: We evaluated clinical outcome and prognostic factors for overall survival of patients with ASCP in comparison to patients with PDAC recorded between 2000 and 2019 in 17 population-based clinical cancer registries at certified cancer centers within the Association of German Tumor Centers (ADT). Results: We identified 278 (0.5%) patients with ASCP in the entire cohort of 52,518 patients with pancreatic cancer. Significantly, more patients underwent surgical resection in the cohort of ASCP patients in comparison to patients with PDAC (*p* < 0.001). In the cohort of 142 surgically resected patients with ASCP, the majority of patients was treated by pancreatoduodenectomy (44.4%). However, compared to the cohort of PDAC patients, significantly more patients underwent distal pancreatectomy (*p* < 0.001), suggesting that a significantly higher proportion of ASCP tumors was located in the pancreatic body/tail. ASCPs were significantly more often poorly differentiated (G3) (*p* < 0.001) and blood vessel invasion (V1) was detected more frequently (*p* = 0.01) in comparison with PDAC. Median overall survival was 6.13 months (95% CI 5.20–7.06) for ASCP and 8.10 months (95% CI 7.93–8.22) for PDAC patients, respectively (*p* = 0.094). However, when comparing only those patients who underwent surgical resection, overall survival of ASCP patients was significantly shorter (11.80; 95% CI 8.20–15.40 months) compared to PDAC patients (16.17; 95% CI 15.78–16.55 months) (*p* = 0.007). ASCP was a highly significant prognostic factor for overall survival in univariable regression analysis (*p* = 0.007) as well as in multivariable Cox regression analysis (HR 1.303; 95% CI 1.013–1.677; *p* = 0.039). Conclusions: In conclusion, ASCP showed poorer differentiation and higher frequency of blood vessel invasion indicative of a more aggressive tumor biology. ASCP was a significant prognostic factor for overall survival in a multivariable analysis. Overall survival of resected ASCP patients was significantly shorter compared to resected PDAC patients. However, surgical resection still improved survival significantly.

## 1. Introduction

Pancreatic cancer is currently the fourth most common cause of cancer-related deaths. However, it is predicted to be the second leading cause of cancer-related mortality by the year 2030 [1]. The most frequent histological subtype of malignancies of the exocrine pancreas is the common pancreatic ductal adenocarcinoma (PDAC) [2]. Several studies described specific molecular subtypes of PDAC [3,4,5]. The comparison of subtypes based on transcriptional profiles shows significant overlap. The quasimesenchymal subtype by Collisson et al. [3], the basal subtype by Moffitt et al. [4] and the squamous subtype by Bailey et al. [5] significantly overlap and have worse clinical outcomes compared to other subtypes [6,7].

Adenosquamous carcinoma of the pancreas (ASCP) is a rare histopathological subtype of PDAC with an incidence of 0.5% to 5% among exocrine pancreatic malignancies [8,9], which consists of both squamous cell carcinoma and ductal adenocarcinoma components [10,11,12]. According to the WHO classification of tumors of the digestive system, squamous differentiation of at least 30% of tumor cells is required for the diagnosis of ASCP [13]. The normal pancreas is histologically devoid of squamous cells [14]. However, squamous metaplasia of the pancreas was found in autopsy studies in 16–48% of cases [15,16]. The pathophysiology of ASCP development remains unclear. Notably, ASCP largely corresponds to the transcriptional PDAC subtypes defined by Bailey as “squamous” and by Collisson as “quasi-mesenchymal” [17].

Clinical data on ASCP is sparse and mostly based on case reports or small series. Similar to patients with common PDAC, ASCP patients commonly present with advanced stage due to late clinical symptoms, which are similar for both tumor entities [18]. The clinical relevance of the distinction of ASCP from PDAC remains controversial. A number of studies suggest that survival of ASCP patients is poor compared to PDAC [19,20,21]. An analysis of surgically resected ASCP and PDAC patients recorded in the Surveillance, Epidemiology, and End Results (SEER) database reported 1-year and 2-year survival rates of 50.7% and 29%, respectively, and a median survival of 12 months for patients with ASCP. In comparison, 1-year and 2-year survival rates for PDAC patients were 60.1% and 35.8%, respectively, and the median survival was 16 months [19]. Other studies suggest that overall survival of ASCP and PDAC patients is similar [18,22].

Surgical resection is the only curative treatment option for pancreatic cancer. However, only 10–20% of patients present with resectable tumors at the time of diagnosis whereas all other patients present with borderline resectable, locally advanced or metastatic disease [23]. The overall 5-year survival rate of pancreatic cancer patients remains as low as 5–8% [24]. Systemic combination chemotherapies with either FOLFIRINOX or gemcitabine plus nab-paclitaxel are still the most important treatment regimens for patients with advanced disease [23]. However, the identification of clinically relevant alterations in individual pancreatic cancers revealed promising targets for personalized targeted treatment strategies such as PARP inhibition for *BRCA 1/2* mutated tumors, NTRK inhibition for NTRK fusion-positive tumors or PD-1 inhibition for microsatellite instability-high (MSI-H) tumors [25].

To date, no specific clinical guidelines for the treatment of ASCP are established. Some authors recommend analogous multidisciplinary treatment protocols developed for PDAC [18]. Surgical resection represents the best therapeutic approach for ASCP patients in whom R0 resection can be achieved [18]. In addition, a single institution case series from the Mayo clinic showed that ASCP patients with R1 resection still had a better survival as compared to those without surgery [26]. The benefit of neoadjuvant and adjuvant chemotherapy for ASCP patients is less clear. Wild et al. reported a retrospective series of 62 ASCP patients, of whom 14 patients were treated with platinum-based chemotherapy in the adjuvant setting. The adjuvantly treated patients had an overall median survival of 19.1 months as opposed to 10.7 months for those without adjuvant treatment [27].

Published data addressing the clinical outcome and prognosis of ASCP patients in comparison to PDAC patients is sparse and somewhat contradictory; therefore, we evaluated the pooled data of the Association of German Tumor Centers (ADT). Herein, all patients with ASCP and common PDAC treated at 17 certified German cancer centers between 2000 and 2019 were analyzed for treatment regimens, clinical outcome and prognostic factors for overall survival.

## 2. Materials and Methods

### 2.1. Study Population

This registry-based study was approved by the institutional ethics committee of the University of Lübeck (# 20-319) and carried out according to the data use regulations of the ADT. The Association of German Tumor Centers (ADT) is the head organization coordinating the nation-wide German Cancer Registry Group, which collects nationwide data from clinical cancer registries for analysis. A minimal data set for various tumor entities is established [28]. This minimal data set includes patient demographics (e.g., age, sex), diagnosis and histology (according to the ICD-O classification [29]), TNM classification [30], treatment (surgery, chemotherapy, radiotherapy) and survival. The database was queried for patients with pancreatic cancers (ICD-O-3: C25.0–C25.9) from 17 centers documented between 2000 and 2019. The cases were further selected for histological diagnosis of ASCP (ICD-O-3 morphology code: 8560/3) and common PDAC, not otherwise specified (PDAC-NOS, ICD-O-3 morphology code: 8500/3).

### 2.2. Study Parameters

The following parameters were selected for analysis: age at diagnosis, sex, histopathological parameters (pT stage (pT0–pT4), pN stage (pN0, pN1, pN2), lymph vessel invasion (L0, L1), vascular invasion (V0, V1), grading (G1–G4) and resection status (R0, R1, R2)). Treatment regimens included: no treatment reported, surgical resection only, neoadjuvant treatment and resection, resection and adjuvant treatment, (radio-)chemotherapy only, neoadjuvant treatment without resection and palliative surgery with (radio-) chemotherapy. Adjuvant therapy included chemotherapy and/or radiotherapy.

Surgical procedures included pancreatoduodenectomy, distal pancreatectomy, total pancreatectomy or other procedures. Chemotherapy regimens comprised gemcitabine-based, fluoropyrimidine-based, platin-based or other regimens. Distant metastases were categorized as M0, M1 or Mx and considered positive (M1) whenever either pathologic pM1 or clinical cM1 was given. Follow up included time of follow up (in months) after diagnosis and status at last follow up (alive or dead).

Grading was dichotomized as G1/2 versus G3/4 and resection status as R0 versus R1/R2 (R+). Due to several updates of the TNM classification between 2000 and 2019 (4th to 8th edition), T2 and T3 were categorized as T2/3 and N1 and N2 were categorized as N+, respectively.

### 2.3. Statistics and Survival Analysis

Data processing and statistical analysis were performed using R (R Foundation for Statistical Computing, Vienna, Austria, version 4.1.2 [31]) and SPSS (IBM, Armonk, NY, USA, version 26.0.0.0). For descriptive statistics median/interquartile range and absolute numbers/percentage of total were used for continuous variables and categorical variables, respectively. Chi-square testing was used to analyze the dependence of two dichotomous variables, and dependence of continuous variables was tested using univariable logistic regression.

Univariable analysis for histology, sex, age, distant metastasis, T-stage, lymph node invasion, lymph vessel invasion, blood vessel invasion, resection margin status, grading and adjuvant treatment was performed. Subsequently, multivariable analysis was done with the same variables. For multivariable analysis, the T-stage was limited to T2/3 versus T4.

The Kaplan–Meier method and Log-Rank test were applied in the univariable survival analysis, and Cox regression in the multivariable survival analysis. The significance level was set to *p* < 0.05 (two sided) and confidence intervals (CI) are reported as 95% CI.

## 3. Results

### 3.1. Patient Cohort and Baseline Characteristics

A total of 52,518 patients with a malignant neoplasm of the pancreas were documented between 2000 and 2019. Of these, 278 (0.5%) patients were diagnosed with an ASCP and 37,941 patients with common PDAC. Median age of the ASCP cohort was 70 years (range: 35–94 years) at time of diagnosis. There were 155 male (55.8%) and 123 female (44.2%) patients. In total, 142 (51.0%) ASCP patients underwent surgical resection; 4 (1.4%) patients received neoadjuvant treatment before resection, 90 (32.4%) were resected only and 38 (13.7%) patients received adjuvant (radio-)chemotherapy after resection. (Radio-)chemotherapy alone was given to 38 (13.7%) patients. Six (2.2%) patients were treated with neoadjuvant intent, but did not undergo surgical resection. No ASCP patient underwent palliative surgery. No specific treatment was reported for 92 (33.1%) ASCP patients. Comparative analyses of the ASCP and PDAC cohort are shown in Table 1A.

A significantly higher proportion of patients underwent surgical resection in the cohort of ASCP patients in comparison to patients with PDAC (*p* < 0.001). In the cohort of 142 surgically resected ASCP patients, the majority of patients was treated by a partial pancreatoduodenectomy (*n* = 63, 44.4%) (Table 1B). However, compared to the cohort of PDAC patients, a significantly higher proportion of ASCP patients underwent distal pancreatectomy (*p* < 0.001). The tumor stage (T) and lymph node status (N) did not differ significantly from the cohort of PDAC patients. However, ASCPs were significantly more often poorly differentiated (G3) in comparison with PDAC (*p* < 0.001). Blood vessel invasion (V1) was detected significantly more often on histopathological examination in ASCP compared to PDAC patients (*p* = 0.01).

### 3.2. Overall Survival of ASCP Compared to PDAC Patients

Overall survival after initial diagnosis was reported for 260 ASCP patients (93.5%). Median time of follow-up was 6.77 months (range: 0–96.4 months) and 211 (75.9%) ASCP patients had died at last follow-up. For the entire study cohort, median overall survival was 6.13 months (95% CI 5.20–7.06) versus 8.10 months (95% CI 7.93–8.22) for ASCP versus PDAC patients, respectively (*p* = 0.094) (Figure 1A). The one-year survival rate was 29.5% and 36.7%, 2-year survival rate was 14.1% and 16.6% and 5-year survival rate was 5.8% and 4.5% for ASCP and PDAC patients, respectively.

Median overall survival of resected ASCP patients (11.80; 95% CI 8.20–15.40 months) was significantly longer compared with unresected ASPC patients (4.30; 95% CI 3.54–5.06 months) (Figure 1B). The same was true for PDAC patients in our study cohort (Appendix A). When comparing only those patients who underwent surgical resection, overall survival of ASCP patients was significantly shorter (11.80; 95% CI 8.20–15.40 months) compared to PDAC patients (16.17; 95% CI 15.78–16.55 months) (*p* = 0.007) (Figure 1C). The one-year survival rate was 49.5% and 61.3%, 2-year survival rate was 23.3% and 34.3% and 5-year survival rate was 10.2% and 10.9% for surgically resected ASCP and PDAC patients, respectively.

### 3.3. Prognostic Factors for Overall Survival of Resected ASCP and PDAC Patients

Given the differences in survival seen between ASCP and PDAC, detailed analysis was performed to identify prognostic factors in resected patients.

In the univariable regression analysis, distant metastasis (M1: 6.03 (3.85–8.21) months; M0: 13.43 (7.22–19.65) months; *p* = 0.002) (Figure 1D) and blood vessel invasion (V1: 8.10 (95% CI 4.89–11.31) months; V0: 17.63 (95% CI 13.43–21.84) months; *p* = 0.031) (Figure 1E) were negative predictive factors for ASCP patients (Table 2a,b). Adjuvant treatment after surgical resection was a positive predictive factor (adjuvant treatment: 22.37 (14.48–30.25) months; no adjuvant treatment: 6.03 (4.56–7.51) months; *p* < 0.001) for ASCP patients (Figure 1F). All these prognostic factors were highly significant for PDAC patients as well (Table 2a,b). However, factors like T-stage, lymph node metastasis, lymph vessel invasion, resection margin status and histopathological grading that were highly significant in univariable analysis for PDAC patients did not reach significance for ASCP patients.

In the multivariable Cox regression analysis, only distant metastasis (HR: 3.436 (95% CI 1.354–8.856); *p* = 0.010) and adjuvant therapy (HR: 5.361 (95% CI 2.858–10.055); *p* < 0.001) remained independent prognostic factors for ASCP patients, whereas multiple other factors were independently prognostic for PDAC patients (Table 3 and Appendix A).

Analyzing the entire study cohort of resected ASCP and PDAC cases, ASCP was a highly significant negative prognostic factor in the univariable (*p* = 0.007) (Appendix A) as well as multivariable Cox regression analysis (HR 1.303; 95% CI 1.013–1.677; *p* = 0.039) (Table 3).

## 4. Discussion

ASCP has been described as a rare but aggressive subtype of pancreatic cancer. The pathophysiology of ASCP development remains unclear. Mainly three hypotheses have been proposed: (i) ductal cells undergo squamous metaplasia due to chronic inflammation caused by chronic pancreatitis or obstruction [32], (ii) the two histologically distinct malignant cell populations develop independently and subsequently merge [33], and (iii) multipotent precursor cells differentiate to a combination of adenocarcinoma and squamous cell carcinoma [34].

Most clinical data on ASCP is sparse and based on case reports or small series. So far, four large-scale hospital- and population-based studies specifically analyzed characteristics and clinical outcomes of ASCP in comparison to PDAC [18,19,35,36]. Katz et al. reported an analysis of the California Cancer Registry from 2000–2007 [18]. The study cohort included 95 ASCP and 14,746 PDAC patients. ASCP patients were resected more frequently in comparison to PDAC. Median overall survival of patients with resected ASCP was 12 months and comparable with PDAC patients. Boyd et al. reported a population-based analysis on 415 ASCP patients and 45,693 PDAC patients extracted from the SEER database between 1988 and 2007 [19]. Compared to PDAC, ASCP was more likely to occur in the pancreatic tail. ASCP tumors were more frequently poorly differentiated, larger, and node positive. Median survival of ASCP patients was 12 months in comparison to 16 months for PDAC patients. Long-term survival following surgical resection was also significantly worse for ASCP. However, surgical resection was the strongest predictor of survival. Hester et al. analyzed the National Cancer Database from 2004 to 2012 and identified 1745 ASCP patients and 205,328 PDAC patients [35]. This analysis did not reveal a significant difference in overall survival between PDAC and ASCP patients (6.2 months and 5.7 months, respectively). However, subgroup analysis of only the resected patients showed a significantly worse overall survival in ASCP patients in comparison to PDAC patients (14.8 months and 20.5 months, respectively). In line with Katz et al. and Boyd et al., ASCP tumors were larger, located more frequently in the pancreatic body/tail and were poorly differentiated. Most recently, Kaiser et al. published a retrospective single-center analysis of patients undergoing surgery for ASCP and PDAC between 2001 and 2017 [36]. The reported cohort comprised 91 ASCP and 3918 PDAC patients. Median overall survival after surgical resection was shorter in ASCP compared to PDAC patients with (10.8 and 20.5 months in PDAC, respectively. However, 5-year survival rates were comparable between both groups. ASCP tumors were larger, more frequently involved lymph nodes, showed poorer differentiation and were located in the pancreatic tail more often.

These above mentioned data are in line with our analysis of the nation-wide pooled cancer registry data. The incidence rate of ASCP among all included pancreatic cancer patients was 0.5%. Significantly more patients underwent surgical resection in our cohort of ASCP patients in comparison to patients with PDAC (*p* < 0.001). A histopathological workup is required for diagnosis of ASCP, and this might represent a selection bias of patients who received surgical resection. In our cohort, 142 patients underwent surgical resection and the majority of these patients was treated by a partial pancreatoduodenectomy (44.4%). Hence, the tumor was most often localized in the pancreatic head. However, compared to the cohort of PDAC patients, significantly more patients underwent distal pancreatectomy (*p* < 0.001) suggesting that a significantly higher proportion of ASCP tumors was located in the pancreatic body/tail. In agreement with previously reported risk factors, ASCPs were significantly more often poorly differentiated (G3) (*p* < 0.001) and blood vessel invasion (V1) was detected more frequently (*p* = 0.01) in comparison with PDAC in our cohort. Median overall survival was 6.13 months (95% CI 5.20–7.06) for ASCP and 8.10 months (95% CI 7.93–8.22) for PDAC patients, respectively (*p* = 0.094). However, when comparing only those patients who underwent surgical resection, overall survival of ASCP patients was significantly shorter (11.80; 95% CI 8.20–15.40 months) compared to PDAC patients (16.17; 95% CI 15.78–16.55 months) (*p* = 0.007). Strikingly, histopathologic distinction of ASCP and PDAC was a highly significant prognostic factor for overall survival in univariable regression analysis (*p* = 0.007) as well as in the multivariable Cox regression analysis (HR 1.303; 95% CI 1.013–1.677; *p* = 0.039).

To the best of our knowledge, this is the first population-based analysis of cancer registry data outside the United States on ASCP in comparison to PDAC. Clinical cancer registries facilitate to gain valuable insights to the real-world situation of epidemiology, treatment and survival of specific subtypes of pancreatic cancers such as ASCP. However, a limitation of registry-based studies include missing data which need to be handled with caution. No treatment was reported in our study cohort of ASCP and PDAC patients for 33.1% and 42.0%, respectively. This can be interpreted in mainly two different ways: (i) either these patients did not undergo any specific cancer treatment and presumably received supportive care or (ii) registry data are constrained by reporting biases. We interpreted those patients as having received best supportive care due to several reasons [37]. First, reporting specific cancer associated treatment data to the cancer registries is mandatory by law in the Federal Republic of Germany. Second, median overall survival in the subgroup of patients with no reported specific cancer treatment compared to those with reported treatment was significantly shorter at only 3.47 (95% CI 2.35–4.59) compared to 8.13 (95% CI 6.74–9.53) and 4.1 (95% CI 3.97–4.23) compared to 11.3 (95% CI 11.10–11.50) months for ASCP and PDAC patients, respectively (Appendix A). This also suggests the interpretation of best supportive care. However, the oncological baseline dataset of the ADT registry does not infer underlying reasons. This group of patients who do not undergo surgical resection might represent failure to provide surgical therapy [38] and therefore should gain further attention in future studies.

In our patient cohort, 13.7% and 23.5% of ASCP and PDAC patients, respectively, were treated by (radio)chemotherapy without surgical resection. In our ASCP cohort, 33.8% of patients were treated with adjuvant therapy after surgical resection, whereas 39.5% of PDAC patients were treated with adjuvant therapy. Katz et al. reported adjuvant therapy for 68% of the resected ASCP patients [18], and Hester et al. reported adjuvant treatment for 62.4% of ASCP and 64.3% of PDAC patients [35]. The reason for this rather low rate of adjuvant treatment in our patient cohort cannot be explained within the scope of the registry data. As other studies previously reported, the retrospective study design limits conclusions on multimodal treatment strategies. The underlying registry-based dataset does not include detailed data on treatment scheme, timing and completion of adjuvant treatment regimens. To date, there are no specific clinical guidelines for treatment of ASCP and treatment according to multidisciplinary treatment protocols that have been developed for PDAC [18]. Although the present and previously published studies clearly show a survival benefit after surgical resection, the benefit of neoadjuvant and adjuvant chemotherapy for ASCP patients is less clear. Wild et al. reported a retrospective series of 62 ASCP patients, of whom 14 patients were treated with platinum chemotherapy in the adjuvant setting. The adjuvant-treated patients had an overall median survival of 19.1 months as opposed to 10.7 months for those who did not [27]. In line, Katz et al. reported a significantly longer median overall survival for ASCP patients with adjuvant therapy. Our data also suggests a significantly better survival for patients with adjuvant treatment in comparison to those patients undergoing surgical resection only. In agreement with previously published studies, we conclude that ASCP patients should be treated in a multimodal setting of surgical resection and (neo-)adjuvant (radio-)chemotherapy. However, the optimal specific treatment protocol needs to be evaluated cautiously.

Over the last decade, several studies described specific molecular PDAC subtypes [3,4,5]. It was shown that ASCP correlates with the quasimesenchymal subtype of PDAC [17]. The quasimesenchymal subtype by Collisson et al. [3], the basal subtype by Moffitt et al. [4] and the squamous subtype by Bailey et al. [5] significantly overlap and have worse clinical outcomes compared to other subtypes [6,7]. Lenkiewiscz et al. recently published a comprehensive study on the genomic landscape of ASCP [39]. They showed in a combined copy number variation and exome sequencing analysis that ASCP genomes contain the common lesions seen in PDACs including *KRAS* and *TP53* mutations, homozygous deletions of *CDKN2A* and *SMAD4* and amplification of *MYC*. Active chromatin at the *SMYD2* locus and increased expression of *RORC* distinguished ASCP from PDAC. Therefore, the authors propose that ASCP evolves from the same lineage as PDAC but consists of enriched levels of *RORC*-positive cancer stem cells which may drive other tumors with adenosquamous features. These data are in line with the interpretation that ASCP are a clinical correlate of the quasimesenchymal PDAC subtype.

Preliminary evidence suggests that this tumor subtypes are more resistant to 5-FU based chemotherapies and alternative drug combinations such as gemcitabine-based regimens are more effective [40]. However, we did not find significant differences in the chemotherapeutic approaches compared to PDAC in our patient cohort (Table 1). Potentially better responses to gemcitabine-based therapies should be considered in ASCP treatment. Thus, further research should be directed to the specific tumor biology of ASCP to identify molecular targets that might allow personalized treatment. A deeper understanding of the underlying tumor biology of ASCP in comparison to PDAC might allow for modifying current treatment strategies. Nevertheless, surgical resection improved survival significantly in our study cohort and should be taken into consideration in a multimodal treatment setting.

## 5. Conclusions

In conclusion, we present the first registry-based analysis of treatment, histopathological features and clinical outcome of ASCP patients in comparison to PDAC patients based on the pooled data of clinical cancer registries by the ADT. The present study shows a poorer differentiation and higher frequency of blood vessel invasion which may be indicative of a more aggressive tumor biology as described previously. Histopathologic distinction of ASCP and PDAC was a significant prognostic factor for overall survival on multivariable analysis. Nevertheless, significantly more ASCP patients underwent only surgical resection without adjuvant treatment in comparison to PDAC patients. Overall survival of resected ASCP patients was significantly shorter compared to resected PDAC patients.

## Figures and Tables

**Figure 1 cancers-14-03946-f001:**
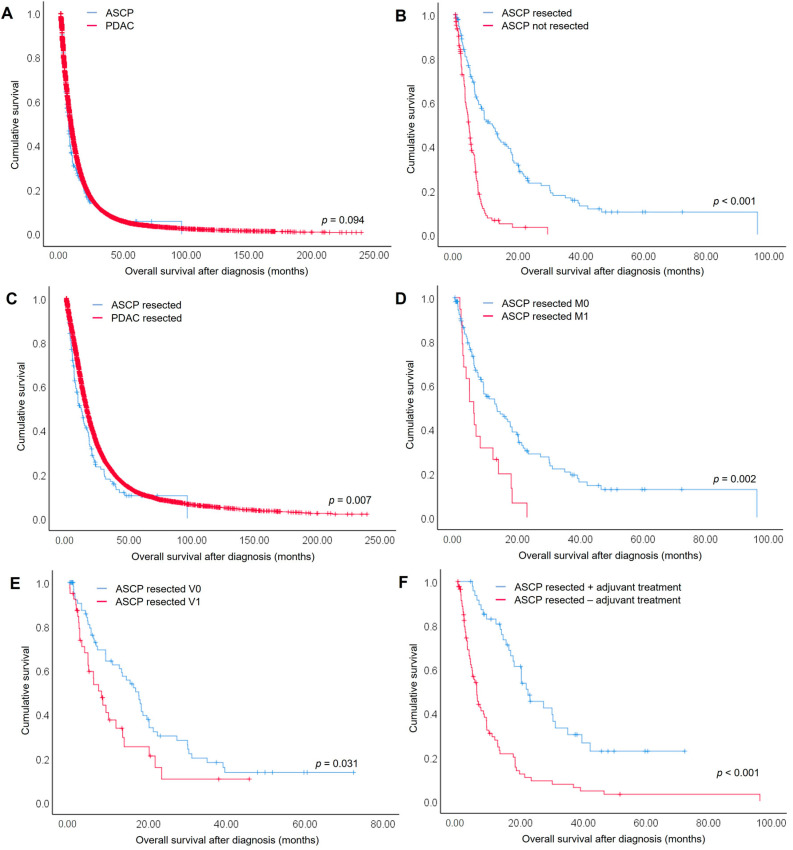
Survival curves of (**A**) all ASCP in comparison to all PDAC patients, (**B**) resected in comparison to not resected ASCP patients, (**C**) resected ASCP in comparison to resected PDAC patients, (**D**) resected ASCP patients without distant metastasis (M0) in comparison to resected ASCP patients with distant metastasis (M1), (**E**) resected ASCP patients without blood vessel invasion (V0) in comparison to resected ASCP patients with blood vessel invasion (V1), and (**F**) resected ASCP patients with adjuvant treatment in comparison to resected patients without adjuvant treatment.

**Table 1 cancers-14-03946-t001:** Baseline characteristics and descriptive statistics of the ASCP and PDAC patient cohorts. (**A**) all included ASCP and PDAC patients and (**B**) cohort of surgically resected ASCP and PDAC patients. ASCP—adenosquamous carcinoma of the pancreas; BSC—best supportive care; N—number; PDAC—pancreatic ductal adenocarcinoma.

Parameter		N (% of Total)/Median (Min–Max)		*p*-Value
		ASCP	PDAC	
A	All Patients			
Total number		278 (100)	37,941 (100)	
Sex				
male		155 (55.8)	20,226 (53.3)	0.416
female		123 (44.2)	17,712 (46.7)	
missing		0 (0)	3 (0.008)	
Median age (years)		70 (35–94)	70 (17–102)	0.385
Distant metastases				
M0		135 (48.6)	12,964 (34.2)	<0.001
M1		88 (31.7)	15,177 (40)	
Mx		55 (19.8)	9800 (25.8)	
Treatment				
No treatment reported; BSC		92 (33.1)	15,919 (42.0)	0.003
Neoadjuvant treatment + resection		4 (1.4)	360 (0.9)	0.402
Resection		90 (32.4)	7370 (19.4)	<0.001
Resection + adjuvant treatment		48 (17.2)	5038 (13.3)	0.051
(Radio-)chemotherapy		38 (13.7)	8928 (23.5)	<0.001
Neoadjuvant treatment without resection		6 (2.2)	274 (0.7)	0.005
Palliative surgery + (radio-)chemotherapy		0 (0)	52 (0.1)	0.537
**B**	**Surgically Resected Patients**			
Total number		142 (100)	12,768 (100)	
Surgical procedure				
Partial pancreatoduodenectomy		63 (44.4)	6297 (49.3)	0.240
Distal pancreatectomy		33 (23.2)	1211 (9.5)	<0.001
Total pancreatectomy		10 (7.0)	900 (7.0)	0.998
Other		36 (25.4)	4360 (34.1)	
Pathological classification of resected tumor				
Tumor stage				
	pT0	0 (0)	23 (0.2)	0.072
	pT1	0 (0)	433 (3.4)	
	pT2/T3	134 (94.4)	10,969 (85.9)	
	pT4	7 (4.9)	926 (7.3)	
	pTx	1 (0.7)	417 (3.3)	
Lymph node status				
	pN0	44 (31.0)	6868 (30.3)	0.984
	pN+	97 (68.3)	8558 (67.0)	
	pNx	1 (0.7)	342 (2.7)	
Grading				
	G1	0 (0)	510 (4.0)	<0.001
	G2	40 (28.2)	5895 (46.2)	
	G3	95 (66.9)	5201 (40.7)	
	G4	0 (0)	62 (0.5)	
	Gx	7 (4.9)	1100 (8.6)	
Lymphatic vessel invasion				
	L0	46 (32.4)	3361 (26.3)	0.885
	L1	67 (47.2)	5034 (39.4)	
	Lx	29 (20.4)	4373 (34.2)	
Blood vessel invasion				
	V0	72 (50.7)	6086 (47.7)	0.010
	V1	40 (28.2)	2037 (16.0)	
	Vx	30 (21.1)	4645 (36.4)	
Resection status				
	R0	77 (54.2)	6278 (49.2)	0.724
	R1	24 (16.9)	2191 (17.2)	
	R2	4 (2.8)	241 (1.9)	
	Rx	37 (26.1)	4058 (31.8)	
Distant metastasis				
	M0	117 (82.4)	10,101 (79.1)	0.734
	M1	21 (14.8)	1966 (15.4)	
	Mx	4 (2.8)	701 (5.5)	
Adjuvant chemotherapy				
Patients treated with adjuvant chemotherapy		48 (33.8)	5038 (39.5)	0.170
Gemcitabine-based		28 (19.7)	3193 (25.0)	
Platin-based		5 (3.5)	225 (1.8)	
Fluoropyrimidine-based		6 (4.2)	466 (3.6)	
Other		9 (6.3)	1154 (9.0)	

**Table 2 cancers-14-03946-t002:** (**a**) Survival analysis for demographic and clinical characteristics of ASCP and PDAC patients. (**b**) Survival analysis for histopathological characteristics of ASCP and PDAC patients. ASCP—adenosquamous carcinoma of the pancreas; CI—confidence interval; N—number; PDAC—pancreatic ductal adenocarcinoma.

(a)								
	ASCP				PDAC			
Parameter	N	Deaths	Median Survival in Months (95% CI)	*p*-Value	N	Deaths	Median Survival in Months (95% CI)	*p*-Value
Overall	134	100	11.80 (8.20–15.40)		11,965	9136	16.17 (15.78–16.55)	
Sex								
male	76	61	9.16 (3.21–15.13)	0.998	6374	4900	15.47 (14.99–15.95)	0.051
female	58	39	12.73 (7.48–17.98)		5591	4236	16.63 (16.07–17.20)	
Age (years)								
≤65	49	36	13.23 (3.75–22.72)	0.497	4686	3517	18.40 (17.66–19.14)	<0.001
>65	85	64	9.17 (3.57–14.77)		7279	5619	14.43 (13.99–14.88)	
Distant metastasis								
M0	111	78	13.43 (7.22–19.65)	0.002	9499	6962	18.40 (17.89–18.91)	<0.001
M1	19	18	6.03 (3.85–8.21)		1784	1606	7.20 (6.69–7.71)	
Adjuvant Therapy								
Yes	48	30	22.37 (14.48–30.25)	<0.001	4973	3525	21.57 (20.92–22.22)	<0.001
No	86	70	6.03 (4.56–7.51)		6992	5611	12.17 (11.71–12.62)	
**(b)**								
	**ASCP**				**PDAC**			
**Parameter**	**N**	**Deaths**	**Median Survival in Months (95% CI)**	***p*-Value**	**N**	**Deaths**	**Median Survival in Months (95% CI)**	***p*-Value**
T-stage								
pT0	0	0			23	7	56.53 (0.00–126.69)	<0.001
pT1	0	0			406	259	34.17 (28.38–39.95)	
pT2/3	127	95	12.17 (8.60–15.73)	0.979	10,303	7877	16.33 (15.92–16.75)	
pT4	6	4	6.07 (1.23–10.91)		829	737	8.90 (8.15–9.65)	
Lymph node metastases								
N0	40	28	14.43 (8.38–20.49)	0.064	3610	2491	22.20 (21.18–23.22)	<0.001
N+	93	71	9.17 (5.01–13.32)		8020	6371	14.23 (13.82–14.64)	
Lymph vessel invasion								
L0	45	29	18.67 (15.85–21.48)	0.096	3252	1975	22.83 (21.79–23.88)	<0.001
L1	66	48	9.23 (4.26–14.20)		4850	3642	15.87 (15.30–16.43)	
Blood vessel invasion								
V0	70	49	17.63 (13.43–21.84)	0.031	5871	3947	19.93 (19.29–20.58)	<0.001
V1	40	28	8.10 (4.89–11.306)		1958	1440	13.47 (12.70–14.24)	
Resection margin status								
R0	75	53	13.50 (9.97–17.03)	0.369	6089	4162	21.00 (20.31–21.69)	<0.001
R+	28	19	9.17 (0.00–28.13)		2355	1793	13.23 (12.61–13.85)	
Grading								
G1/G2	38	27	11.80 (2.26–21.34)	0.508	6021	4397	19.87 (19.21–20.52)	<0.001
G3/G4	89	66	10.90 (4.50–17.30)		4943	3956	12.60 (12.16–13.04)	

**Table 3 cancers-14-03946-t003:** Multivariable Cox regression analysis for survival of resected ASCP and PDAC patients. ASCP—adenosquamous carcinoma of the pancreas; CI—confidence interval; HR—hazard ratio; PDAC—pancreatic ductal adenocarcinoma.

	ASCP		PDAC		ASCP + PDAC	
Parameter	HR (95% CI)	*p*-Value	HR (95% CI)	*p*-Value	HR (95% CI)	*p*-Value
Histology (ASCP vs. PDAC)					1.303 (1.013–1.677)	0.039
Distant metastases (M1 vs. M0)	3.436 (1.354–8.856)	0.010	1.614 (1.453–1.793)	<0.001	1.625 (1.464–1.804)	<0.001
Adjuvant therapy (no vs. yes)	5.361 (2.858–10.055)	<0.001	1.871 (1.758–1.990)	<0.001	1.894 (1.781–2.014)	<0.001

## Data Availability

Data were obtained from the Association of German Tumor Centers (Arbeitsgemeinschaft Deutscher Tumorzentren, ADT) and are available from the authors with the permission of the Association of German Tumor Centers.

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
