# Peer review of "Clinical Outcome and Prognostic Factors of Pancreatic Adenosquamous Carcinoma Compared to Ductal Adenocarcinoma—Results from the German Cancer Registry Group"

_cancers, 2022, doi:10.3390/cancers14163946_

Round 1

Reviewer 1 Report

Braun et al. have addressed our points and we believe the paper has reached the quality of the publication.

Reviewer 2 Report

Dear Authors,

thank you for the comments and additions. I have no further remarks.

This manuscript is a resubmission of an earlier submission. The following is a list of the peer review reports and author responses from that submission.

Round 1

Reviewer 1 Report

An interesting study exploring a timely, important topic in pancreatic cancer.

The identification of prognostic factors in this setting is of pivotal importance, and it is very interesting how the authors compared some features of ASCP with those of PDAC.

Some revisions are necessary in my opinion:

- a linguistic revision would be recommended;

- the introduction section should be expanded, and the background of medical treatment for pancreatic cancer better explained, also adding some recent papers discussing this topic (PMID: 34358103 ; PMID: 32593337)

- Table 2 and Table 3 should be divided into four tables to help the readability of the paper.

Some changes are required.

Reviewer 2 Report

To the authors:

Adenosquamous carcinoma of the pancreas (ASCP) was a rare histopathological subtype of pancreatic ductal adenocarcinoma and its clinicopathological characteristics is poorly understood. Dr. Braun R and colleagues queried the nation-wide pooled cancer registry data and reported clinical outcomes of ASCP. They found that ASCP was associated with poorer differentiation and higher frequency of blood vessel invasion indicative of a more aggressive tumor biology. In multivariable analysis, ASCP was a significant prognostic factor for overall survival. Given adjuvant treatment after surgical resection was a positive predictive factor (adjuvant treatment: 22.37 months; no adjuvant treatment: 6.03 months; p<0.001) for ASCP patients, the authors concluded that ASCP patients should be treated in a multimodal setting of surgical resection and adjuvant chemotherapy.

This paper is very informative regarding ASCP, a rare histopathological subtype of pancreatic ductal adenocarcinoma with an incidence of 0.5%, and will be very useful in the clinical practice when encountering this rare disease. The beauty of this paper is also including the outcomes of the ASCP patients who didn’t undergo surgery. Seeing the relatively shorter median observation period of ASCP not-resected in the figure, this can be thanks to the development of diagnostic modalities, especially EUS-FNA, I would assume. I think this paper should be accepted for publication in this journal after all my concerns have been appropriately addressed by the authors in the revision.

In order to improve the quality of the paper, I would like to ask the following questions.

1.           It is important to have statisticians in the authors’ contribution to ensure that the nation-wide registry data are used correctly, and the results are not misleading. These current analyses need to be reviewed by the experts. And it’s better to have their names as co-authors.

2.           From my point of view, it may be better to explain the HR of the worse prognostic factors with more than 1.0. For example, the HR of ASCP should be 1.304 instead of 0.767 and the reference should be PDAC instead of ASCP, and the HR of adjuvant therapy should be 0.528 instead of 1.894 and the reference should be “no” instead of “yes”, (same for age).

Reviewer 3 Report

In the manuscript titled “Clinical outcome and prognostic factors of pancreatic adenosquamous carcinoma compared to ductal adenocarcinoma – results from the German cancer registry group” Braun et al. present a study on clinical outcome and prognostic factors of patients diagnosed with pancreatic adenosquamous carcinoma in German cohort. Among 52,518 patients with pancreatic cancer 278 (0,5%) patients had adenosquamous carcinoma. Authors showed that pancreatic adenosquamous carcinoma has more aggressive tumor biology and survival of patients is shorter when compared to PDAC.

It‘s an interesting study on a rare histological subtype of pancreatic cancer. However, there are still few points for improvement.

Points for improvement

1.     Quality of the tables must be improved. Especially Table 2 is difficult to read

2.     Abbreviation “PDAC-NOS” (line 188) needs explanation

3.     In Your opinion, what should be the direction of future research in patients diagnosed with pancreatic adenosquamous carcinoma? Is there any place for the improvement of the treatment? Some thoughts might be added to the discussion.
